# The New Era of Three-Dimensional Histoarchitecture of the Human Endometrium

**DOI:** 10.3390/jpm11080713

**Published:** 2021-07-25

**Authors:** Manako Yamaguchi, Kosuke Yoshihara, Nozomi Yachida, Kazuaki Suda, Ryo Tamura, Tatsuya Ishiguro, Takayuki Enomoto

**Affiliations:** Department of Obstetrics and Gynecology, Niigata University Graduate School of Medical and Dental Sciences, Niigata 951-8510, Japan; manako0131@med.niigata-u.ac.jp (M.Y.); nyachida@med.niigata-u.ac.jp (N.Y.); sudakazuaki@med.niigata-u.ac.jp (K.S.); ryo-h19@med.niigata-u.ac.jp (R.T.); tishigur@med.niigata-u.ac.jp (T.I.); enomoto@med.niigata-u.ac.jp (T.E.)

**Keywords:** endometrium, endometrial gland, endometrial-related disease, three-dimensional (3D), 3D imaging, tissue clearing, histology

## Abstract

The histology of the endometrium has traditionally been established by observation of two-dimensional (2D) pathological sections. However, because human endometrial glands exhibit coiling and branching morphology, it is extremely difficult to obtain an entire image of the glands by 2D observation. In recent years, the development of three-dimensional (3D) reconstruction of serial pathological sections by computer and whole-mount imaging technology using tissue clearing methods with high-resolution fluorescence microscopy has enabled us to observe the 3D histoarchitecture of tissues. As a result, 3D imaging has revealed that human endometrial glands form a plexus network in the basalis, similar to the rhizome of grass, whereas mouse uterine glands are single branched tubular glands. This review summarizes the relevant literature on the 3D structure of mouse and human endometrium and discusses the significance of the rhizome structure in the human endometrium and the expected role of understanding the 3D tissue structure in future applications to systems biology.

## 1. Introduction

Our knowledge of the histology and pathology of the human body is primarily based on the observation of two-dimensional (2D) thin sections. However, analyses of only a few selected tissue sections may be insufficient to understand the complex organization of organs. To overcome the limitations of 2D studies, several researchers have developed various techniques. For example, digital reconstruction techniques of serial pathological section images have been applied to the three-dimensional (3D) visualization of tissue structures, such as vessels, glands, tumors, and volume rendering [1,2,3,4,5]. Tissue clearing techniques combined with optical sectioning and confocal microscopy or light-sheet microscopy (LSM) have enabled whole-organ and whole-body imaging with single-cell resolution [6,7,8,9,10,11]. Using recent advances in those techniques, 3D structures of various normal and diseased tissues, such as mouse brain [12,13], human pancreas and pancreatic cancer [2,14], human lung adenocarcinoma [5], human nonalcoholic fatty liver disease [15], and human gingiva [16]), have been elucidated. Particularly, 3D imaging with various fluorescent labels can provide system-level approaches that can be used to study cellular circuits in organisms and improve the spatial understanding of structural and functional cellular and subcellular information in complex mammalian bodies and large human specimens [7,10]. Furthermore, the combination of 3D tissue and omics data will contribute to the elucidation of tissue physiology and disease pathophysiology in the future.

The human endometrium is a dynamic tissue that undergoes cyclic shedding, regeneration, and differentiation during a woman’s reproductive years. The glandular epithelium is a highly complex tissue that contains undulations and bifurcations; this tissue can also change its morphology from moment to moment under the influence of hormones [17,18]. The human endometrium is divided into a basal layer, which remains during menstruation, and a functional layer, which is sloughed during menstruation. It is believed that repeated regeneration of the endometrium occurs as a result of stem/progenitor cells located in the basal layer [19,20,21]. In classical histology, the endometrium is classified as a single branched tubular gland, similar to the stomach and Brunner’s glands of the small intestine [22,23], and the glands of the basalis have been described as blind ends [24,25,26,27]. However, recent 3D observations have revealed that the structure of adult human endometrial glands is more complex [28,29].

In this review, we present an overview of previous reports on the 3D structure of the mouse and human endometrium and introduce applications of 3D histoarchitecture to systems biology and discuss future prospects.

## 2. 3D Structure of the Endometrium

Several studies on the 3D structure of the endometrium have been reported in both mice and humans [26,28,29,30,31,32]. The subjects, methods, and findings of the significant studies are shown in Table 1.

BABB: Benzyl alcohol/benzyl benzoate. BABB is comprised of two parts benzyl benzoate and one part benzyl alcohol [13]. CUBIC: Clear, unobstructed brain/body imaging cocktails and computational analysis [9]. The updated CUBIC protocol IV use two solutions, CUBIC-L (mixture of 10 wt% N-butyldiethanolamine and 10 wt% Triton X-100) and CUBIC-R+(mixture of 45 wt% 2,3-dimethyl-1-phenyl-5-pyrazolone, 30 wt% nicotinamide, and 5 wt% Nbutyldiethanolamine) [29].

### 2.1. Mice

Generally, the rodent uterus is easy to handle due to its small size, and the uterine glands are described as simple tubular structures rather than as tightly coiled or extensively branched structures, as in humans and domestic animals [17,33]. Therefore, the observation and understanding of the 3D structure of mouse uterine glands are relatively simple. Significant studies on the 3D structure of mouse uterine glands have used whole-mount immunofluorescence and tissue clearing methods [30,31,32]. Optical sections of whole-mount organs are advantageous because the tissue does not need to be dismantled and can be converted to a 3D image on a computer after; the cross-section of the organ in any plane can be computed.

Goad et al. performed imaging of thick uterine sections and whole uteri collected from prepubertal TCF/Lef:H2B-GFP transgenic mice and revealed that the uterine glands are simple tubes with branches directly connected to the luminal epithelium and are only present toward the anti-mesometrial side of the uterus [30]. In addition, the development of uterine glands and high Wnt signaling activity was shown to be mainly restricted to the anti-mesometrial side of the uterus, which suggests that Wnt signaling is active in uterine adenogenesis and that a differential gradient of Wnt proteins is responsible for attracting blastocysts to the anti-mesometrial side of the uterus [30].

Vue et al. performed volumetric imaging and generated a 3D model of the uterine glands within the developing postnatal mouse uterus from P0 to P21 [31]. They proposed a new 3D staging system for uterine gland morphology that classifies the process of gland development from the luminal epithelium of the uterus into five stages (Stage 1: bud; Stage 2: teardrop; Stage 3: elongated; Stage 4: sinuous; and Stage 5: primary branches) [31]. The endings of individual glands were described as blind ends.

Arora et al. used a tissue clearing technique combined with confocal imaging to identify and quantify the dynamic changes in the structure of the mouse uterine lumen in preparation for implantation [32]. They also succeeded in visualizing 3D slit-like structures termed uterine crypts, where mouse embryos are implanted [32].

Thus, all previous reports have shown that the 3D structure of mouse uterine glands consists of simple tubular glands that are continuous from the luminal epithelium of the uterus, often with branching and terminating in a blind end. Since transgenic models are available in mice, the combination of fluorescent labeling with confocal microscopy and LSM has the advantage of following the spatiotemporal changes of labeled molecules. The application of 3D imaging technology is expected to aid in the investigation of the role of various signaling pathways in the development of the mouse endometrium, as well as in implantation and to reveal their contribution to the spatial uterine environment [32]. However, since humans experience a physiological phenomenon termed menstruation, the use of mice, which do not menstruate, as a model for endometrial regeneration, development of endometrial-related diseases, and implantation of fertilized eggs has its limitations. Recently, the spiny mouse, which has a naturally occurring menstrual cycle, was discovered and characterized as a viable model sharing many attributes of physiological menstruation with humans [34]. This newly discovered menstruating rodent enables us to investigate the influence of menstruation in a 3D structure of normal uterine endometrium. Moreover, the use of other mouse models that mimic menstruation, pregnancy, and other menstrual disorders may be useful to understand dynamic changes of the 3D endometrial structure.

### 2.2. Humans

Humans are rare among mammals in that they experience menstruation [35]. Approximately 100 years ago, women experienced menstruation about 40 times in their lifetimes due to amenorrhea during pregnancy and lactation. However, in developed countries today, the number of menstrual periods experienced in a lifetime may be as high as 400 due to women’s social advancement and lifestyle changes [36,37]. Although menstruation is essential for human reproduction, it is also associated with diseases such as abnormal uterine bleeding, dysmenorrhea, endometriosis, and premenstrual syndrome, which significantly impact women’s physical, mental, and social health. In recent years, improving menstrual health literacy has become a challenge among other issues related to women’s health, but one obstacle is that basic uterine and menstrual physiology are still not fully understood [37]. A proper understanding of the actual 3D structure of the human endometrium is vital for the further progression of basic endometrial research.

In the 1980s, Schmidt et al. attempted to depict the 3D configuration of human endometrial glands. They used formalin-fixed samples sliced into 1-mm-thick sections that were then stained with hematoxylin, degreased, and cleared in toluene. They drew the morphology of the endometrial glands by hand using a camera lucida, and the morphology was depicted as a single branched tubular gland, just as in classical histology textbooks [38]. In the 1990s and 2000s, several studies reported the observation of the 3D structure of the endometrium based on the computerized 3D reconstruction of serial histology sections. However, at that time, images were limited to showing fragmentary 3D structures of a small portion of the cylindrical glands and the surrounding blood vessels [39,40,41].

Manconi et al. used multiphoton excitation microscopy to observe the microvascular and glandular structures of the human endometrium and to perform 3D imaging [26]. However, the maximum depth of observation was 120 µm, and they could not depict the glandular structures in all layers from the superficial to the basal layers [26].

In 2016, Arora et al. used tissue clearing and confocal microscopy to successfully observe not only the mouse uterus but also the full thickness of biopsied human endometrium in the proliferative phase [32]. They noted that “3D renderings of endometrial lumen and glands revealed increased glandular complexity of the human uterus compared with mouse”, but they did not analyze the detailed structure of human endometrial glands [32].

Two recent studies reported unique 3D structures of human endometrial glands that defy previous characterizations [28,29]. Tempest et al. reconstructed 100 consecutive 4-µm-thick sections from seven human endometrial samples (two proliferative, three secretory, and two postmenopausal) using computer software, after which they selected several glands and visualized them in 3D [28]. All sections included the full thickness of the endometrium from the lumen to the endometrial-myometrial junction, but because of their reconstructed 3D thickness of 400 µm, only a portion of the glands (either functionalis portion or basalis portion) were separately depicted three-dimensionally, and the entire image of each individual gland from the basal layer to the lumen was not shown. By joining the 2D images of the endometrial functionalis/basalis junction from the subendometrial myometrium of some of the glands to reconstruct 3D images, they noted that the deep basalis glands branch intricately and run horizontally along the myometrium, forming a root-like network [28].

We recently reported tissue-clearing-based 3D histoarchitecture with fluorescence staining of full-thickness human uterine endometrial tissues [29]. In this study, we applied our updated clear, unobstructed brain/body imaging cocktails and computational analysis (CUBIC) protocol IV [9], and succeeded in substantially clearing approximately 200–1500 cubic mm of normal human uterine tissues (four proliferative, seven secretory, and nine menstrual) and visualizing detailed 3D structures of endometrial glands from the basal layer to the lumen using LSM. We performed a 3D reconstruction of proliferative and secretory samples and revealed a horizontally expanding plexus morphology of the endometrial glands near the bottom of the endometrium (Figure 1).

When we analyzed the morphology of the glands individually, we found that 68% of the glands were branched and that 90% of the branched points were located in the lower third of the endometrium. Furthermore, we discovered that some endometrial glands shared a plexus and rose toward the luminal epithelium (Figure 2, Appendix A).

In other words, the human endometrial glands have a unique and complex 3D structure distinct from the previously assumed single branched tubular gland structure. We then generated 3D reconstructions of menstrual samples to investigate whether the plexus network corresponded to the basalis, which does not detach during menstruation. We found that the plexus network remained behind in all menstrual samples observed, which demonstrates that this is a major component of the basalis (Figure 3, Appendix A). Although previous scanning electron microscope studies have shown that the endometrium sheds in a piecemeal manner with denuded and intact sections next to each other [42], our study has demonstrated that the functional layer of the endometrium sheds off completely during menstruation, probably due to a limited area where we assessed 3D structure [29].

We named the plexus network of the basal gland the “rhizome”, after comparing it to an erosion-resistant rhizomatous plant such as grass. We then created a database containing 3D histologic images of human uterine endometrium Available online: https://true.med.niigata-u.ac.jp/ (accessed on 29 May 2020).

## 3. Significance of the Rhizome Structure in Human Basal Glands

Both we and Tempest et al. considered that there might be structural advantages to rhizomes in the human endometrium, which undergoes cyclic shedding and regeneration [28,29]. The human endometrium has great regenerative ability after menstruation, childbirth, or artificial scraping, which is thought to be due to the presence of stem/progenitor cells in the basalis [19,20,21]. In the field of botany, rhizomatous plants, which have stems running horizontally underground, are known to be highly resistant to natural and artificial erosion [43,44]. Rhizomes in the human endometrium may also be more efficient than crypts in preserving epithelial stem/progenitor cells from any shedding. The absence of rhizomes in the uterine glands of the mouse, a mammal without menstruation, also suggests a link between the physiological phenomenon of cyclic shedding and regeneration and rhizome structure [30,31,32]. In the future, it is desirable to confirm the 3D structure of the uterine gland in domestic animals other than mice and in primates that experience menstruation. In addition, the 3D structure of the human endometrium was previously observed in adult samples [28,29]. The development of human uterine glands begins during fetal growth, continues after birth, and is completed before puberty [17]. It is, therefore, necessary to observe premenstrual and pubertal uterine glands to determine the point at which rhizome structures are formed.

## 4. Application of 3D Histoarchitecture to Systems Biology and Future Prospects

Several studies indicate that human endometrial epithelial glands are monoclonal in origin [20,21,45], which suggests that they arise from a single stem/progenitor cell. In particular, our group and others have recently reported the clonal expansion of cancer-associated mutations such as *PIK3CA*, *KRAS*, and *PTEN* in histologically normal endometrial glands using single gland sequencing or laser capture microdissection [46,47]. Furthermore, endometrial intraepithelial neoplasia, as part of a newer classification of hyperplastic lesions, is defined as the monoclonal proliferation of architecturally and cytologically altered premalignant endometrial glands, which are prone to transformation to endometrioid carcinoma [48]. These results provide an important insight into the molecular aspects of endometrial carcinogenesis with regard to the “stem cell-hit” theory; if genetic alterations occur in stem/progenitor cells in the basalis, they must be consistent and be passed onto daughter cells through subsequent clonal expansion, ultimately resulting in the clonal occupation of whole glands [19]. By understanding the conventional 2D shape of the gland, this theory can be used to understand how one gland obtains genetic alterations. However, it is not sufficient to explain how mutated cells within glands increase and expand in the region. If an endometrial gland has a monoclonal composition based on the rhizome structure, we can put forward a new theory: if genetic alterations occur in stem/progenitor cells in the rhizome, those alterations spread to several glands that share the rhizome, and mutant glands then expand in the region as the mutated rhizome enlarges the area in the basalis.

On the contrary, Tempest et al. reported the existence of epithelial multipotent stem cells with the ability to regenerate the entire endometrial gland element in humans based on the analysis of cytochrome c oxidase (CCO) gene mutations in mitochondrial DNA (mtDNA), using single-cell capture microdissection. They also showed that CCO-deficient and CCO wild-type cells might be mixed in a single vertical gland by analyzing CCO mutations in mtDNA, and they speculated that basal glands generated from two types of stem cells merge to form a single gland [28]. Whether the confluence of different ancestral glands could also occur requires further investigation. Future studies on human endometrial genomics and stem/progenitor cells should consider the rhizome structure of the basalis.

Recently, it has been reported that cancer-associated gene mutations, such as those in *KRAS* and *PIK3CA*, are present in cases of endometriosis and adenomyosis, which are benign endometrium-related diseases [47,49,50,51]. Inoue et al. reported the presence of oligoclonality and recurrent KRAS mutations in uterine adenomyosis tissues [51]. They indicated that uterine adenomyosis might arise from the ectopic proliferation of mutated epithelial cell clones, as variant allele frequencies (VAFs) were higher in adenomyosis lesion epithelium recovered by LMD and targeted deep sequencing compared with bulk samples. In addition, they found that VAFs for mutations encoding oncogenic *KRAS* p. G12/G13 alterations were significantly increased in the normal endometrium of adenomyosis patients compared with cases with neither adenomyosis nor endometriosis, which suggests that KRAS mutant clones were expanding in the normal endometrium of these adenomyosis patients. They considered that *KRAS*-mutated adenomyosis clones originate from the eutopic normal endometrium [51]. Our previous study succeeded in visualizing 3D adenomyosis tissues using the tissue clearing method and found that eutopic normal endometrial glands directly invaded the myometrium and that ectopic glands formed ant colony-like networks with complex branching in the myometrium [29]. In the future, it is expected that the pathogenic mechanism of endometrial diseases will be further clarified by combining 3D images of diseased tissue with genomic analysis.

In a previous study, the largest human endometrial sample that we observed in 3D was approximately 15 square mm [29]. Although we performed multiple sampling in some cases, we could not observe the entire uterus [29]. Recently, Zhao et al. developed a new tissue permeabilization method for large intact human organs [11]. If the entire human uterus can be made transparent and its structure observed, more knowledge will be gained about differences in the 3D structure of the endometrium depending on the site (e.g., fundus, corpus, isthmus, and cervix) and the relationship between the paths of blood vessels and the 3D glandular structure.

Tissue clearing and 3D imaging technology with single-cell resolution combined with 3D immunohistochemistry can allow visualization of cellular connectivity and dynamics and can contribute to the basis of organ- or organism-level systems biology [7]. However, because in vivo genetic labeling and fluorescent dye tracing are not applicable in the human sample study [52], cellular and molecular analysis of human organs requires a post-sampling staining method based on diffusion penetration using fluorescently labeled antibodies [9] or uterine perfusion model [53]. Since antigenicity is dependent on histological preparation conditions (e.g., fixation and clearing), the detailed histological preparation conditions should be described for each working antibody [8]. In addition, the uterine perfusion model may be suitable for the assessment of vasculature because the intravascular blood clots interfere with the staining of the vascular system [53]. By overcoming these challenges, we hope to elucidate the localization of stem/progenitor cells in the human endometrium and differences in hormone sensitivity of glandular epithelial cells during the menstrual cycle or in the basalis/functionalis at single-cell resolution in 3D. Further determination of the antibodies suitable for 3D staining of centimeter-sized human tissues is necessary for future cellular and molecular analyses.

## 5. Conclusions

The recent development of 3D imaging technology has revealed the rhizome structure of human endometrial glands in the basalis. A future challenge is to combine 3D imaging with omics and molecular analyses. The new 3D histoarchitecture of the human uterine endometrium and endometrial diseases has excellent potential to serve as a fundamental resource to develop various fields, including pathology, pathophysiology, reproduction, and oncology.

## Figures and Tables

**Figure 1 jpm-11-00713-f001:**
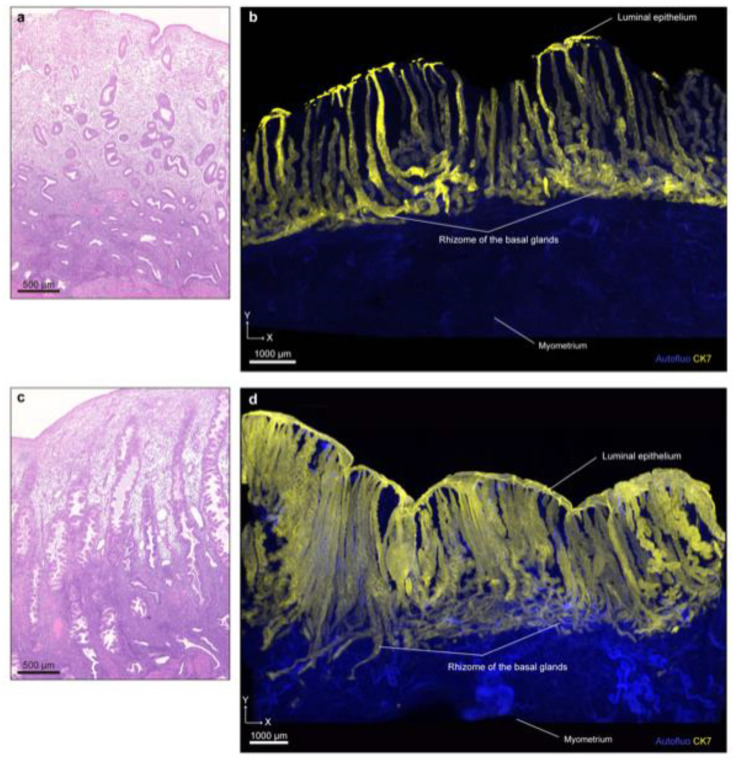
3D imaging of human uterine tissue using CUBIC. (**a**) Hematoxylin and eosin (H&E)-stained image of formalin-fixed paraffin-embedded (FFPE) endometrial tissue in the proliferative phase (age 46). (**b**) Reconstructed XY section (z = 500 µm) of the same sample shown in panel (**a**) after clearing by CUBIC. (**c**) H&E-stained image of FFPE endometrial tissue in the secretory phase (age 46). (**d**) Reconstructed XY section (z = 1000 µm) of the same sample shown in panel (**c**) after clearing by CUBIC. Images were obtained by LSF microscopy. Autofluorescence was measured by excitation at 488 nm. CK7-expressing endometrial epithelial cells were measured by excitation at 532 nm. Autofluo, autofluorescence; CK7, cytokeratin 7.

**Figure 2 jpm-11-00713-f002:**
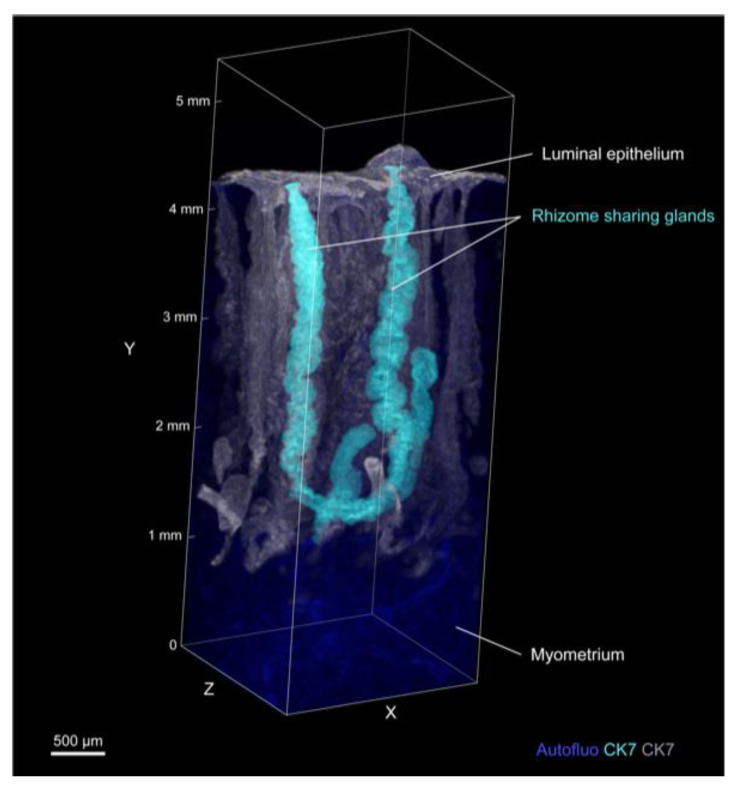
3D morphology of the rhizome sharing glands (age 46, secretory phase). Three-dimensional reconstruction of the distribution of branched glands, which were pseudocolored and separated as new channels by the Surface module in Imaris [29]. Images were obtained by LSF microscopy. Autofluorescence was measured by excitation at 488 nm. CK7-expressing endometrial epithelial cells were measured by excitation at 532 nm. Autofluo, autofluorescence; CK7, cytokeratin 7. See also Appendix A.

**Figure 3 jpm-11-00713-f003:**
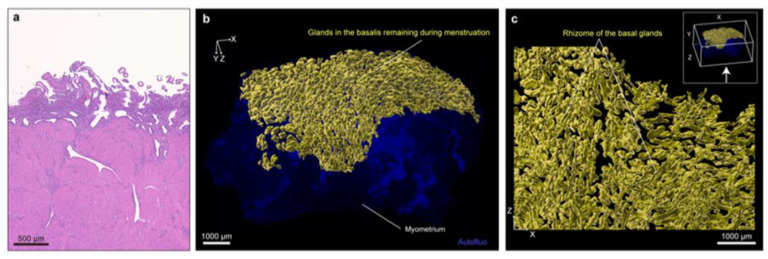
3D morphology of endometrial glands in a case during menstruation (age 45). (**a**) H&E-stained image of FFPE endometrial tissue in a menstrual sample. (**b**) Reconstructed 3D image of the same sample shown in panel (**a**). Yellow object: glands in the basalis remaining during menstruation. (**c**) Extended XZ-plane view (view from the direction of the arrow in the upper right inset) of the same sample shown in panel (**a**). The yellow object shows the rhizome structure of the basal glands remaining during menstruation. Yellow objects were constructed by the Surface module in Imaris [29]. After surface extraction, each structure was manually curated, and extra surface signals were eliminated. Images were obtained by LSF microscopy. Autofluorescence was measured by excitation at 488 nm. Autofluo, autofluorescence. See also Appendix A.

**Table 1 jpm-11-00713-t001:** The significant studies of the 3D structure of mouse and human endometrium.

Study (Year of Publication)	Subjects	Methods	3D Observation Range	Findings
Goad et al. [30] (2017)	Prepubertal mouse	Tissue clearing (ScaleA2), stereoscope	Whole uterus	Mouse uterine glands are simple tubes with branches directly connected to the luminal epithelium and are only present towards the antimesometrial side of the uterus.
Vue et al. [31] (2018)	Prepubertal mouse	Tissue clearing (ScaleA2), light-sheet microscopy	Whole uterus	3D models of uterine glands within the developing postnatal mouse uterus
Arora et al. [32] (2016)	Adult mouse, Human	Tissue clearing (BABB), confocal microscope	Mouse: whole uterusHuman: full-thickness of the endometrium	Mouse: Visualization and characterization of changes in the mouse endometrium from fertilization through implantation. Human: Glandular complexity of the human uterus was increased compared with the mouse.
Manconi et al. [26] (2003)	Human	Human endometrial sections of 100 µm thickness, multiphoton excitation microscopy	Part of the endometrium	The surface epithelium consists of a monolayer of cuboidal cells that lines the uterine lumen and is in continuity with the glandular epithelial cells. The endometrial glands appear to have both simple and branched tubular shapes.
Tempest et al. [28] (2020)	Human	Digital reconstruction of serial pathological section images (4 µm × 100 slices)	3D visualization of part of the glands(Pathological sections included full-thickness of the endometrium)	The deeper basalis glands demonstrated a complex, often branched organization, enveloping one another horizontally in a mycelium/root-like configuration on the underlying myometrium.
Yamaguchi et al. [29] (2021)	Human	Tissue clearing (updated CUBIC protocol IV) of normal human uterine tissues (about 5–15 mm square), light-sheet microscopy	Full-thickness of the endometrium	The basal glands formed the rhizome structure, that is, a horizontally expanding plexus morphology. Some glands shared the rhizome with other glands.

## Data Availability

Not applicable.

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
