# Peer review of "The New Era of Three-Dimensional Histoarchitecture of the Human Endometrium"

_jpm, 2021, doi:10.3390/jpm11080713_

Round 1

Reviewer 1 Report

This is a well-written paper and a pleasure to read.

Remarks on manuscript jpm-1302634 “The new era of three-dimensional histoarchitecture of the human endometrium” by Yamaguchi et al.

The review by Yamaguchi et al. is well written and comprehensive in its coverage of recent developments in microscopy, tissue clearing and methodology to enable whole-organ imaging of architectural features such as the glands within human endometrium. This review follows up on their original work published earlier this year in iScience, ref #29 in this manuscript. While that paper was more concerned with their own findings, it is nice to see a broader picture painted in this review, with the work of others, e.g., Tempest et al., mentioned to full merit. The authors could include more on adenomyosis as recently presented at ESHRE but this is not the focus of the paper, of course.

It was nice to read this paper; and I only have a few suggestions to make it even nicer:

  1. Line 60: “… to systems biology and future prospects.” Include “discuss” before “future prospects”, to make sense: “… to systems biology, and discuss future prospects.”

  1. Table 1 – please include the year of publication in your ‘study’ column, to give the reader an idea of how far apart these studies were, or how close together.

  1. Line 102: “Since mice are capable of gene transfer, …” is somewhat prone to misunderstandings. Rewrite as “Since transgenic models are available in mice, …”.

  1. Line 108: “On the contrary, since humans experience a unique physiological phenomenon termed menstruation, the use of mice, which do not menstruate, as a model for endometrial regeneration, development of endometrial-related diseases, and implantation of fertilized eggs has limitations.” Better to start with “However, …” instead of “On the contrary, …”, as the latter can mean the other side of an argument. Note that menstruation is not unique to humans. Insert “its” before “limitations”.

  1. Could the authors include a reference to the (menstruating!) spiny mouse (Acomys cahirinus) here, or elsewhere in the manuscript?

  1. Line 114: This should read “Humans are rare among mammals in that they experience menstruation [34].”

  1. Line 124: Insert “actual”, to make clear what is meant: “A proper understanding of the actual 3D structure …”

  1. Line 140: Leave out “the … method”: “In 2016, Arora et al. used tissue clearing and confocal microscopy…” – also in line 272.

  1. Line 152: “3D-ized” is not a word. Replace with “depicted three-dimensionally”.

  1. Line 162: Square mm (mm2) are a two-dimensional parameter. How much material was used, in cubic mm (mm3)? This occurs again in line 279.

  1. Figure 2: Would it be possible to include an arrow measuring the distance from top to bottom of the specimen? I know that there is a scale bar, but it would be helpful – and very impressive.

  1. Line 189: “classified” is not the word to use here. Perhaps rephrase as “… distinct from the previously assumed single branched tubular structure.” Leave out “that is” before “distinct”.

  1. Line 192: Should read “We found that the plexus network remained behind in all menstrual samples observed, which demonstrates that this is a major component of the basalis …”.

  1. Line 295: “we can expect” – replace with “we hope”.

  1. Line 302: Should read: “The recent development of 3D imaging technology has revealed the rhizome structure of human endometrial glands in the basalis.”

  1. Regarding the paragraph beginning in line 286, are the authors aware of perfusion methods, e.g., as described by Stirland et al., 2015, PMID26184049? These might enable labelling of vasculature or tissue within the uterus over time.

Author Response

Reviewer 1

Comments and Suggestions for Authors

This is a well-written paper and a pleasure to read. 

Remarks on manuscript jpm-1302634 “The new era of three-dimensional histoarchitecture of the human endometrium” by Yamaguchi et al.

The review by Yamaguchi et al. is well written and comprehensive in its coverage of recent developments in microscopy, tissue clearing and methodology to enable whole-organ imaging of architectural features such as the glands within human endometrium. This review follows up on their original work published earlier this year in iScience, ref #29 in this manuscript. While that paper was more concerned with their own findings, it is nice to see a broader picture painted in this review, with the work of others, e.g., Tempest et al., mentioned to full merit. The authors could include more on adenomyosis as recently presented at ESHRE but this is not the focus of the paper, of course.

It was nice to read this paper; and I only have a few suggestions to make it even nicer:

  1. Line 60: “… to systems biology and future prospects.” Include “discuss” before “future prospects”, to make sense: “… to systems biology, and discuss future prospects.”

According to the reviewer’s comment, we added “discuss” before “future prospects”.

  1. Table 1 – please include the year of publication in your ‘study’ column, to give the reader an idea of how far apart these studies were, or how close together.

According to the reviewer’s comment, we added “the year of publication” into the “study” column of Table 1.

  1. Line 102: “Since mice are capable of gene transfer, …” is somewhat prone to misunderstandings. Rewrite as “Since transgenic models are available in mice, …”.

Thank you for your comment. We rewrote this sentence properly.

  1. Line 108: “On the contrary, since humans experience a unique physiological phenomenon termed menstruation, the use of mice, which do not menstruate, as a model for endometrial regeneration, development of endometrial-related diseases, and implantation of fertilized eggs has limitations.” Better to start with “However, …” instead of “On the contrary, …”, as the latter can mean the other side of an argument. Note that menstruation is not unique to humans. Insert “its” before “limitations”.

Thank you for your comment. We rewrote this sentence according to the reviewer’s comments.

  1. Could the authors include a reference to the (menstruating!) spiny mouse (Acomys cahirinus) here, or elsewhere in the manuscript?

According to the comments from both the reviewer 1 and the reviewer 2, we added a reference about the spiny mouse in the manuscript.

  1. Line 114: This should read “Humans are rare among mammals in that they experience menstruation [34].”

Thank you for your comment. We rewrote this sentence properly.

  1. Line 124: Insert “actual”, to make clear what is meant: “A proper understanding of the actual 3D structure …”

According to the reviewer’s comment, we added “actual” into this sentence.

  1. Line 140: Leave out “the … method”: “In 2016, Arora et al. used tissue clearing and confocal microscopy…” – also in line 272.

According to the reviewer’s comment, we deleted “the … method”.

  1. Line 152: “3D-ized” is not a word. Replace with “depicted three-dimensionally”.

According to the reviewer’s comment, we replace “3D-ized” to “depicted three-dimensionally”.

  1. Line 162: Square mm (mm2) are a two-dimensional parameter. How much material was used, in cubic mm (mm3)? This occurs again in line 279.

We revised the description about material size properly.

  1. Figure 2: Would it be possible to include an arrow measuring the distance from top to bottom of the specimen? I know that there is a scale bar, but it would be helpful – and very impressive.

According to the reviewer’s comment, we added a scale bar into Figure 2.

  1. Line 189: “classified” is not the word to use here. Perhaps rephrase as “… distinct from the previously assumed single branched tubular structure.” Leave out “that is” before “distinct”.

According to the reviewer’s comment, we revised this sentence properly.

  1. Line 192: Should read “We found that the plexus network remained behind in all menstrual samples observed, which demonstrates that this is a major component of the basalis …”.

According to the reviewer’s comment, we revised this sentence properly.

  1. Line 295: “we can expect” – replace with “we hope”.

According to the reviewer’s comment, we replace “can expect” with “hope”.

  1. Line 302: Should read: “The recent development of 3D imaging technology has revealed the rhizome structure of human endometrial glands in the basalis.”

According to the reviewer’s comment, we revised this sentence properly.

  1. Regarding the paragraph beginning in line 286, are the authors aware of perfusion methods, e.g., as described by Stirland et al., 2015, PMID26184049? These might enable labelling of vasculature or tissue within the uterus over time.

We appreciate that the reviewer 1 introduced the perfusion methods to us. We added a description about the perfusion method to the revised manuscript.

Reviewer 2 Report

It's really great to see a review on a new and exciting field of research. As yet there are only a few studies that have used 3D technologies and the authors have described each clearly in each section.

On line 108, the authors note the limitations of using mice to study human endometrium as they don't menstruate. With the recent evidence of the spiny mouse as a rodent that menstruates, could the authors please include this and comment on its future potential. There are also a number of mouse models that mimic menstruation, pregnancy and other menstrual disorders such as Asherman's which should also be mentioned in this paragraph. Whilst interventions can't be done in humans, many labs use old world primates to study menstruation and disorders, their use could be mentioned in the future prospects section.

The supplementary videos are great. It would be beneficial to have included a description of the planes that are moved through during the video so the viewer can clearly follow each viewpoint. e.g.  from above, moving to transversely to longitudinal etc etc.

In video S2 and Figure 3 the authors look at menstrual endometrium.  The video would indicate that the entire basalis is exposed to the lumen. Previous SEM studies have shown that the endometrium sheds in a piecemeal manner, with denuded and intact sections next to each other. Could the authors please comment on whether they saw any evidence of this in their 3D samples?

Author Response

Comments and Suggestions for Authors

It's really great to see a review on a new and exciting field of research. As yet there are only a fewstudies that have used 3D technologies and the authors have described each clearly in each section.

On line 108, the authors note the limitations of using mice to study human endometrium as they don't menstruate. With the recent evidence of the spiny mouse as a rodent that menstruates, could the authors please include this and comment on its future potential.

We referred a paper about the spiny mouse and added several comments on future potential of the spiny mouse.

There are also a number of mouse models that mimic menstruation, pregnancy and other menstrual disorders such as Asherman's which should also be mentioned in this paragraph. Whilst interventions can't be done in humans, many labs use old world primates to study menstruation and disorders, their use could be mentioned in the future prospects section.

According to the reviewer’s comments, we added the achievement of mouse models that mimic menstruation.

The supplementary videos are great. It would be beneficial to have included a description of the planes that are moved through during the video so the viewer can clearly follow each viewpoint. e.g.  from above, moving to transversely to longitudinal etc etc.

According to the reviewer’s comments, we added some descriptions into the movie.

In video S2 and Figure 3 the authors look at menstrual endometrium.  The video would indicate that the entire basalis is exposed to the lumen. Previous SEM studies have shown that the endometrium sheds in a piecemeal manner, with denuded and intact sections next to each other. Could the authors please comment on whether they saw any evidence of this in their 3D samples?

According to the reviewer’s recommendation, we added some comments of menstrual endometrium to the revised manuscript.